# Scientific sinkhole: The pernicious price of formatting

**Allana G. LeBlanc[1]***, **Joel D. Barnes[1]**, **Travis J. Saunders[2]**, **Mark S. Tremblay[1]**, **Jean-Philippe Chaput**  **[1]**

1 Healthy Active Living and Obesity Research Group, CHEO Research Institute, Ottawa, ON, Canada,
2 Department of Applied Human Sciences, University of Prince Edward Island, Charlottetown, PE, Canada

* allanagwleblanc@gmail.com

## Abstract

### Objective

To conduct a time-cost analysis of formatting in scientific publishing.

### Design

International, cross-sectional study (one-time survey).

### Setting

Internet-based self-report survey, live between September 2018 and January 2019.

### Participants

Anyone working in research, science, or academia and who submitted at least one peer-reviewed manuscript for consideration for publication in 2017. Completed surveys were available for 372 participants from 41 countries (60% of respondents were from Canada).

### Main outcome measure

Time (hours) and cost (wage per hour x time) associated with formatting a research paper for publication in a peer-reviewed academic journal.

### Results

The median annual income category was US$61,000–80,999, and the median number of publications formatted per year was four. Manuscripts required a median of two attempts before they were accepted for publication. The median formatting time was 14 hours per manuscript, or 52 hours per person, per year. This resulted in a median calculated cost of US$477 per manuscript or US$1,908 per person, per year.

### Conclusions

To our knowledge, this is the first study to analyze the cost of manuscript formatting in scientific publishing. Our results suggest that scientific formatting represents a loss of 52 hours, costing the equivalent of US$1,908 per researcher per year. These results identify the

**Data Availability Statement:** All relevant data are within the manuscript and its Supporting Information files.

**Funding:** The authors received no specific funding for this work.

**Competing interests:** The authors have declared that no competing interests exist.

hidden and pernicious price associated with scientific publishing and provide evidence to advocate for the elimination of strict formatting guidelines, at least prior to acceptance.

## Introduction

Publishing scientific findings is an important part of the scientific process and dates back to as early as 1665 [1]. Since before this time, scientists have been using the peer-review process to share their findings, document their successes, translate knowledge, and share research across fields of study. Top journals may reach hundreds of thousands of people across the globe whereas lower ranked journals may lead to research gathering virtual dust in a far corner of the internet. Many funding bodies and institutions also require scientists to publish their results in a timely manner in an effort to account for public funds, and to increase transparency in research. However, modern day scientific publishing has become an industry unto itself, with thousands of journals to choose from and each of these journals with its own unique set of formatting requirements.

A reality, as per Chapman and Swade [2], is that over 80% of all scientific papers are rejected at least once prior to publication. Other metrics paint a more depressing picture still. In 2013, *Nature* published less than 8% of all submissions [3]. As such, many scientists may be inclined to submit a rejection letter to reject their rejection to starve off disappointment [2]. Of course, not all studies are novel or of sufficient scientific rigor to warrant publication. But increasingly, scientists are frustrated not only because they have been critiqued on the scientific valour of their work, but because of the time they have lost due to journal-specific, idiosyncratic formatting during the submission process. Many scientists have voiced their concerns of time lost due to formatting on several social media platforms, and via the commentary section of several journals. As argued by Guo (and Moore and Budd thereafter), there is no scientific advantage to employing different referencing formats [3,4,5]. Budd goes on to estimate that time spent re-formatting accounts for 10,000 scientist hours per year, per journal [3]. Elsevier reported that nearly one in three scientists reported "preparing manuscripts" as the work activity they found the most frustrating or time consuming [6].

Interestingly, we were unable to find any previous studies that aimed at quantifying the economic costs of authors' time spent on formatting for scientific publications. To address this knowledge gap, we conducted a survey to estimate the time and wage costs of formatting scientific manuscripts for publication in peer-reviewed journals. We hypothesized that the cost of formatting would be substantial and strategies should be put in place to mitigate this cost while preserving the essence of the scientific process.

## Materials and methods

### Study protocol and data collection

A self-report electronic survey containing 10 questions was sent initially through the authors' scientific affiliations and networks via a snowball method to individuals who may be responsible for submitting scientific papers for publication in peer-reviewed journals (S1 Appendix). For the purpose of this work, formatting was defined as total time related to formatting the body of the manuscript, figures, tables, supplementary files, and references. Respondents were asked not to count time spent on statistical analysis, writing, or editing. The survey was promoted through email (e.g., personal email and organizational list serves), websites and blogs (e.g., www.haloresearch.ca, Obesity Panacea), social media (i.e., Twitter, Facebook) and word-

of-mouth. The survey was live between October 2018 and January 2019. Due to the snowball methodology, it is not possible to determine a response rate, but 85% of surveys that were started, were completed. Ethics approval was obtained from the Research Ethics Board at the Children's Hospital of Eastern Ontario Research Institute. By completing the survey respondents provided their consent. The collection method complied with the terms and conditions for the websites from which participants were recruited. Data were collected and analyzed anonymously. The survey took approximately 5 minutes to complete and all information gathered was kept confidential with no personal identifiers retained.

## Study population

Participants were required to have access to the internet to complete the survey. Only participants who indicated that they had submitted at least one peer-reviewed paper in the previous year (2017) were eligible to complete the survey. The survey was only offered in English. There was no minimum or maximum age requirement to participate.

## Outcomes

The primary outcome was the time (hours) and wage cost (wage rate per hour x time) associated with formatting scientific papers. Time spent formatting was the total amount of time taken to format a manuscript for publication in a peer-reviewed academic journal. Participants were asked to include the time it took them to format the paper for initial submission, as well as any additional time required until the paper was accepted and considered ready for publication. The wage cost of formatting was calculated using annual income to estimate wage rates per hour (11 income categories, see S1 Appendix). Occupation was used to estimate annual income in cases of refusal responses; however, almost all participants (99%) disclosed their annual income. Occupation was used for descriptive statistics and subgroup analysis.

## Statistical analysis

This was the first study we could find to calculate the wage cost of formatting in scientific research. As such, our methods and analysis were exploratory. We considered any wage cost associated with formatting to be significant as this is not a cost that is typically accounted for in research funding, grant applications, job descriptions, or considerations for promotion. We summarized the continuous variables by median and median absolute deviation (to account for the over-dispersed and skewed distributions of several of the variables) and categorical variables by frequency and percentage. Participants were asked to convert their gross annual personal income to US dollars (USD) using an online currency converter (www.xe.com/currencyconverter). Several participants responded with an income value in their national currency. These responses were converted to USD using the same currency converter. For participants who did not provide an income category (selected the "Other" response item), it was estimated as the median income category for the occupation group with which they identified. To calculate the wage cost of formatting, we used an estimate of 1950 hours in a working year.

Univariate outlier detection was performed by computing robust z-scores for all count variables related to the outcome variables (number of manuscripts submitted in 2017, time spent formatting a manuscript for submission, time spent formatting a manuscript from acceptance to publication, number of journals submitted to before a manuscript was accepted, time spent re-formatting a manuscript for re-submission). Robust z-scores greater than three were flagged as outliers. An audit was then performed on these flagged values and respondents with responses deemed unreasonably high were removed from the analysis; 20 respondents in total

were removed from the analysis. All analyses were completed using RStudio 1.1.463 (Boston, MA).

## Results

### Study population

The survey resulted in 921 page views and 458 participants completed at least one question of the survey. Complete surveys were available from 419 respondents. Twenty-seven respondents reported that they were not responsible for submitting and/or formatting manuscripts for publication in a peer-reviewed journal and were excluded from analysis. A further 20 respondents were removed as outliers with implausible responses. The final analytic sample included 372 respondents. Table 1 shows the summary demographic characteristics of these participants. The youngest participant was 22 years-old and the oldest participant was 72 years-old, suggesting that there were respondents from across the career spectrum. Participants were from 41 countries representing six continents (60% of which were from Canada), and included 16 clinicians/healthcare providers (e.g., medical doctor, nurse), 257 scientists/researchers (e.g., professor, scientist, post-doctoral fellow), 24 research assistants/managers, 66 students (e.g., undergraduate, masters, doctoral), and 9 people who responded "other".

**Table 1. Demographic information of participants (n = 372).**

| Demographic variable | |
|---|---:|
| **Age (median, MAD)** | 37.0 (10.4) |
| **Gender (n, %)** | |
| **Men** | 165 (44.4) |
| **Women** | 207 (55.6) |
| **Country (n, %)** | |
| **Australia** | 15 (4.0) |
| **Canada** | 224 (60.2) |
| **Spain** | 13 (3.5) |
| **United Kingdom** | 16 (4.3) |
| **United States** | 34 (9.1) |
| **Other** | 70 (18.8) |
| **Occupation (n, %)** | |
| **Clinician/health care provider** | 16 (4.3) |
| **Scientist/researcher** | 257 (69.1) |
| **Research assistant/managers** | 24 (6.5) |
| **Student** | 66 (17.7) |
| **Other** | 9 (2.4) |
| **Salary (n, %)** | |
| **< $21,000 per year** | 37 (9.9) |
| **$21,000–80,999 per year** | 182 (48.9) |
| **$81,000–140,999 per year** | 101 (27.2) |
| **$141,000–200,999 per year** | 26 (7.0) |
| **> = $201,000 per year** | 26 (7.0) |

MAD: median absolute deviation

**Table 2. Outcomes related to cost of formatting for scientific publications.**

| Outcome (median, MAD) | Per manuscript | Per person, per year |
|---|---|---|
| Number of manuscripts responsible for submitting and/or formatting per year | 4 (3.0) | - |
| Number of submissions before publication | 2 (1.5) | - |
| **Hours** | | |
| Time spent on initial formatting | 4 (3.0) | 16 (14.8) |
| Time spent re-formatting for re-submission | 3 (3.0) | 6 (5.9) |
| Total time spent formatting from initial submission until publication | 14 (11.1) | 52 (48.9) |
| **Cost** | | |
| Wage-cost | US$477 | US$1908 |

MAD: median absolute deviation

## Outcomes

Outcome measures are presented in Table 2. Due to positive skewness in the data, all outcome measures are reported as median values. The median annual income category was US$61,000–80,999 and the median number of publications formatted in 2017 was four. Manuscripts required a median of two attempts before they were accepted for publication. Median time spent formatting was 14 hours per manuscript. Median total time for all manuscripts published in 2017 was 52 hours per person per year. This resulted in wage costs of US$477 per manuscript, or US$1,908 per person per year.

Sensitivity analyses of outcome measures by age groups ($\leq$37 years vs. >37 years; median split), gender (men vs. women) and occupation (scientists vs. non-scientists) are reported as Supporting Information (S1–S3 Tables). Overall, the cost of formatting a peer-reviewed scientific publication (per person, per year) was significantly higher in those over 37 years of age (US$3,669 vs. US$938), in men (US$2,015 vs. US$1,726) and among non-scientists (US$2,391 vs. US$947).

## Discussion

To our knowledge, this is the first study to analyze time and wage cost associated with formatting in scientific publishing. Our results suggest that each manuscript costs 14 hours, or US$477 to format for publication in a peer-reviewed journal. This represents a loss of 52 hours or a cost of US$1,908 per person-year. This is an important discovery as currently time for formatting is considered valueless, even in the scientific community [7]. The cost of formatting is similar to the cost of publication fees for many open access journals and suggests that researchers may need to build in this formatting time into grant application budgets, though it is possible that granting agencies may receive this funding request unfavourably.

Alternatively, it is hoped that a growing number of journals will recommend no strict formatting guidelines, at least at first submission but preferably until acceptance, to alleviate the unnecessary burden on scientists. In 2012, Elsevier initiated a process like this in the journal *Free Radical Biology & Medicine* with "Your Paper, Your Way", a simplified submission process with no strict formatting requirements until the paper has been accepted for publication. Others have suggested that journals assume the role of standardizing formatting, an expense that may be justified given publication charge that many impose [8]. Another suggestion may be that journal publishers compensate authors for their time spent formatting (and/or other editorial services). Our analysis suggests the time-cost for formatting may soon be a luxury that few scientists can afford.

As with any study, this work has several strengths and limitations. On a positive note, we were able to recruit a large and diverse sample, suggesting that this is an area of importance and interest within the scientific community. We also had a high proportion of completed surveys, suggesting that participant burden was low and interest level was high. Further, we had representation from several countries around the world, and across various levels of employment, suggesting that our sample may be crudely representative of the scientific community. Among the limitations, the observational nature of the study precludes inferences about causality. We have also not asked respondents about their field of study, and it is likely that many respondents were from health sciences given our field of research and the snowball methodology used. The representation of respondents was also largely from Canada (60%). This work was also based on self-report data and cannot be verified. It is possible that some scientists may over-estimate the time they spend formatting; however, we did try to address this issue during analysis by removing implausible outliers and median values were used for analysis. This survey was also based on a convenience sample and respondent bias is likely. We did not ask participants about their training in copy-editing, referencing software, or graphic design. It is possible that proficiency with word processing software may impact amount of time required to format. Future work may address these shortcomings.

In summary, we estimate the cost of formatting a peer-reviewed scientific publication at US $477 per manuscript, or US$1,908 per person, per year. We hope that this quantification, although suggestive and not definitively conclusive, will bring more attention to the costs associated with formatting for scientific journals and will help to encourage more flexible formatting practices for publications, at least at first submission, as well as more recognition of the burden of formatting by employers and funding agencies.

## Supporting information

**S1 Appendix. Survey questions.**
(DOCX)

**S1 Table. Outcomes related to cost of formatting for scientific publications, by age group.**
(DOCX)

**S2 Table. Outcomes related to cost of formatting for scientific publications, by gender.**
(DOCX)

**S3 Table. Outcomes related to cost of formatting for scientific publications, by occupation.**
(DOCX)

## Acknowledgments

We thank the participants for completing this survey.

## Author Contributions

**Conceptualization:** Allana G. LeBlanc, Joel D. Barnes, Travis J. Saunders, Mark S. Tremblay, Jean-Philippe Chaput.

**Formal analysis:** Joel D. Barnes.

**Methodology:** Allana G. LeBlanc, Joel D. Barnes, Travis J. Saunders, Mark S. Tremblay, Jean-Philippe Chaput.

**Writing – original draft:** Allana G. LeBlanc, Jean-Philippe Chaput.

**Writing – review & editing:** Joel D. Barnes, Travis J. Saunders, Mark S. Tremblay.

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
