## [Decision Letter · Decision Letter 0]

30 Aug 2019

[EXSCINDED]

PONE-D-19-20185

Scientific sinkhole: The pernicious price of formatting

PLOS ONE

Dear Dr. Chaput,

Thank you for submitting your manuscript to PLOS ONE. After careful consideration, we feel that it has merit but does not fully meet PLOS ONE’s publication criteria as it currently stands. Therefore, we invite you to submit a revised version of the manuscript that addresses the points raised during the review process.

We would appreciate receiving your revised manuscript by Oct 14 2019 11:59PM. To enhance the reproducibility of your results, we recommend that if applicable you deposit your laboratory protocols in protocols.io, where a protocol can be assigned its own identifier (DOI) such that it can be cited independently in the future. For instructions see: http://journals.plos.org/plosone/s/submission-guidelines#loc-laboratory-protocols

We look forward to receiving your revised manuscript.

Kind regards,

Alireza Abbasi

Academic Editor

PLOS ONE

Journal Requirements:

Additional Editor Comments (if provided):

Please make sure the limitations of observational studies have been acknowledged, including in the abstract, and provide supports for statements of causation, where applicable.

Reviewers' comments:

Reviewer's Responses to Questions

**Comments to the Author**

1. Is the manuscript technically sound, and do the data support the conclusions?

Reviewer #1: Yes

Reviewer #2: Yes

2. Has the statistical analysis been performed appropriately and rigorously? 

Reviewer #1: Yes

Reviewer #2: Yes

3. Have the authors made all data underlying the findings in their manuscript fully available?

Reviewer #1: Yes

Reviewer #2: Yes

4. Is the manuscript presented in an intelligible fashion and written in standard English?

Reviewer #1: Yes

Reviewer #2: Yes

5. Review Comments to the Author

Reviewer #1: When preparing a manuscript to a journal for publication, it is the fundamental responsibility of the authors to make sure that the written texts are composed to a high standard and free from grammar mistakes. Formatting the manuscript according to the requirement of a particular journal, however, adds little value to the manuscript. A journal has the freedom to define its own image and style, but should not really request the authors to stick to a specific format when submitting an article for review. As Allana G LeBlanc et al have found from their research, scientists spent a significant amount of time each year doing journal specific formatting of their manuscripts. This is transformed directly into extra cost in research. According to the study of Allana G LeBlanc et al, it costs around $1900 per researcher per year performing manuscript formatting. To say that this is the amount of money wasted might sounds a bit harsh, but this certainly could be the money saved for more important things. The study of Allana G LeBlanc et al is well conducted and their finding is reasonably representative.

Reviewer #2: This is a study about an important issue in scholarly communication. It is about the time scholars spend on formatting manuscripts for publications in journals. In spite of its cost implications, the issue hasn’t received enough attention from journals, publishers or researchers who study scholarly communication.

The study has used a short and relatively effective questionnaire survey to ask scientists about the time they spend on the task and their salary. The statistics is ok and median and median absolute deviation that are safe measures in this case have been used and outliers have been removed.

The findings, although suggestive, and not conclusive (this, I believe, should be stated by the authors) can raise the awareness among journals and stimulate more discussion and action.

The authors might be interested in the following piece in relation to the same issue.

Khan, A., Montenegro‐Montero, A., & Mathelier, A. (2018). Put science first and formatting later. EMBO reports, 19(5).

A few issues that I noticed are as below:

• My major problem with the results is that it lacks context. The authors haven’t asked the respondents about their field of study (even broadly like social science, natural science etc.) so we really don’t know what this data represent. The sampling has been snowball mainly and self-selective through website pop-up etc. Given that the authors are from health sciences there is chance that most respondents are from health sciences too and again this should be mentioned.

•

• The respondents, although said to be from 41 countries, are mostly (the majority) from Canada. Even the number from USA, given the number of scientists it has is very low. This should be mentioned in limitation and statements about country of participants should be toned down in abstract etc.

• Another statement that I think needs mending is that in a few places the author mention the survey was available for a short period of time. Surveys are usually open for 10 days or a month or so. This has been open for about 4 months and that is not really a short period of time in survey studies.

• I personally, based on my personal experience, think 14 hours for formatting each paper might be a bit over-estimation, but I am from social science and maybe respondents coming from different fields have different experience.

• In Table 1, the column header for the second column should be (n, %).

• I think providing more analysis can result in more insight. Giving median and MAD for hours spent on formatting by gender, occupation, age group and checking if there are significant differences can provide more context for the data. Also the number of papers each group have said they format a year could be very different.

On a side note, author contribution section has a long story about how the paper was developed with disagreement about formatting, which is unusual because that is not a matter of personal preference and people usually decide on the basis of the journal they want to submit to.

6. PLOS authors have the option to publish the peer review history of their article (what does this mean?). If published, this will include your full peer review and any attached files.

Reviewer #1: Yes: Quanmin Guo

Reviewer #2: No

---

## [Author Response · Author response to Decision Letter 0]

5 Sep 2019

RESPONSE TO REVIEWERS’ COMMENTS

General note: Modifications that have been made to the article as a result of the reviewers’ comments have been highlighted to facilitate their identification. 

Comments to the Authors:

Editor Comments:

Please make sure the limitations of observational studies have been acknowledged and provide supports for statements of causation, where applicable.

Response: The limitations of observational studies have been added to the paper and we modified the language to avoid causation statements. 

Reviewer #1: 

When preparing a manuscript to a journal for publication, it is the fundamental responsibility of the authors to make sure that the written texts are composed to a high standard and free from grammar mistakes. Formatting the manuscript according to the requirement of a particular journal, however, adds little value to the manuscript. A journal has the freedom to define its own image and style, but should not really request the authors to stick to a specific format when submitting an article for review. As Allana G LeBlanc et al have found from their research, scientists spent a significant amount of time each year doing journal specific formatting of their manuscripts. This is transformed directly into extra cost in research. According to the study of Allana G LeBlanc et al, it costs around $1900 per researcher per year performing manuscript formatting. To say that this is the amount of money wasted might sounds a bit harsh, but this certainly could be the money saved for more important things. The study of Allana G LeBlanc et al is well conducted and their finding is reasonably representative.

Response: We thank the reviewer for this positive feedback. 

Reviewer #2: 

This is a study about an important issue in scholarly communication. It is about the time scholars spend on formatting manuscripts for publications in journals. In spite of its cost implications, the issue hasn’t received enough attention from journals, publishers or researchers who study scholarly communication. The study has used a short and relatively effective questionnaire survey to ask scientists about the time they spend on the task and their salary. The statistics is ok and median and median absolute deviation that are safe measures in this case have been used and outliers have been removed. The findings, although suggestive, and not conclusive (this, I believe, should be stated by the authors) can raise the awareness among journals and stimulate more discussion and action.

Response: We thank the reviewer for this feedback. We have added the suggestion about “suggestive and not conclusive” to the manuscript. 

The authors might be interested in the following piece in relation to the same issue.

Khan, A., Montenegro‐Montero, A., & Mathelier, A. (2018). Put science first and formatting later. EMBO reports, 19(5).

Response: We thank the reviewer for this. We have added this paper to our manuscript. 

A few issues that I noticed are as below:

• My major problem with the results is that it lacks context. The authors haven’t asked the respondents about their field of study (even broadly like social science, natural science etc.) so we really don’t know what this data represent. The sampling has been snowball mainly and self-selective through website pop-up etc. Given that the authors are from health sciences there is chance that most respondents are from health sciences too and again this should be mentioned.

Response: We agree with the reviewer. This has been added to the manuscript. 

• The respondents, although said to be from 41 countries, are mostly (the majority) from Canada. Even the number from USA, given the number of scientists it has is very low. This should be mentioned in limitation and statements about country of participants should be toned down in abstract etc.

Response: We agree with the reviewer. We have added this point in the limitations and mentioned in the abstract that 60% of respondents were from Canada. 

• Another statement that I think needs mending is that in a few places the authors mention the survey was available for a short period of time. Surveys are usually open for 10 days or a month or so. This has been open for about 4 months and that is not really a short period of time in survey studies.

Response: This has been removed from the manuscript. 

• I personally, based on my personal experience, think 14 hours for formatting each paper might be a bit over-estimation, but I am from social science and maybe respondents coming from different fields have different experience.

Response: It is difficult to know. These are the data we have and it includes time spent formatting and re-formatting until publication. This is in line with my experience but I agree that it may be field dependent. 

• In Table 1, the column header for the second column should be (n, %).

Response: We decided not to put a unit in the header and instead put it after each variable. For example, age has a different unit (median, MAD) than the other variables (n, %). 

• I think providing more analysis can result in more insight. Giving median and MAD for hours spent on formatting by gender, occupation, age group and checking if there are significant differences can provide more context for the data. Also the number of papers each group have said they format a year could be very different.

Response: We thank the reviewer for this suggestion. Sensitivity analyses by gender, age group and occupation have been added to the manuscript. 

On a side note, author contribution section has a long story about how the paper was developed with disagreement about formatting, which is unusual because that is not a matter of personal preference and people usually decide on the basis of the journal they want to submit to.

Response: We agree with the reviewer. We have shortened this paragraph as requested.

---

## [Decision Letter · Decision Letter 1]

16 Sep 2019

Scientific sinkhole: The pernicious price of formatting

PONE-D-19-20185R1

Dear Dr. Chaput,

We are pleased to inform you that your manuscript has been judged scientifically suitable for publication and will be formally accepted for publication once it complies with all outstanding technical requirements.

With kind regards,

Alireza Abbasi

Academic Editor

PLOS ONE

Additional Editor Comments (optional):

Reviewers' comments:

Reviewer's Responses to Questions

**Comments to the Author**

1. If the authors have adequately addressed your comments raised in a previous round of review and you feel that this manuscript is now acceptable for publication, you may indicate that here to bypass the “Comments to the Author” section, enter your conflict of interest statement in the “Confidential to Editor” section, and submit your "Accept" recommendation.

Reviewer #2: All comments have been addressed

2. Is the manuscript technically sound, and do the data support the conclusions?

Reviewer #2: Yes

3. Has the statistical analysis been performed appropriately and rigorously? 

Reviewer #2: Yes

4. Have the authors made all data underlying the findings in their manuscript fully available?

Reviewer #2: Yes

5. Is the manuscript presented in an intelligible fashion and written in standard English?

Reviewer #2: Yes

6. Review Comments to the Author

Reviewer #2: The revised manuscript is sufficient for publication. The research has some limitations that I mentioned before but they have been mentioned in the manuscript now. The research is overall interesting and useful.

7. PLOS authors have the option to publish the peer review history of their article (what does this mean?). If published, this will include your full peer review and any attached files.

Reviewer #2: Yes: Hamid R. Jamali

---

## [Editor Report · Acceptance letter]

20 Sep 2019

PONE-D-19-20185R1 

Scientific sinkhole: The pernicious price of formatting 

Dear Dr. Chaput:

I am pleased to inform you that your manuscript has been deemed suitable for publication in PLOS ONE. Congratulations! Your manuscript is now with our production department. 

With kind regards,

on behalf of

Dr. Alireza Abbasi 

Academic Editor

PLOS ONE